# How Should Pre-Trained Language Models Be Fine-Tuned Towards Adversarial Robustness?

**Xinshuai Dong**
Nanyang Technological University & Sea AI Lab
dongxinshuai@outlook.com

**Luu Anh Tuan**
Nanyang Technological University
anhtuan.luu@ntu.edu.sg

**Min Lin**
Sea AI Lab
linmin@sea.com

**Shuicheng Yan**
Sea AI Lab
yansc@sea.com

**Hanwang Zhang**
Nanyang Technological University
hanwangzhang@ntu.edu.sg

## Abstract

The fine-tuning of pre-trained language models has a great success in many NLP fields. Yet, it is strikingly vulnerable to adversarial examples, *e.g.*, word substitution attacks using only synonyms can easily fool a BERT-based sentiment analysis model. In this paper, we demonstrate that adversarial training, the prevalent defense technique, does not directly fit a conventional fine-tuning scenario, because it suffers severely from catastrophic forgetting: failing to retain the generic and robust linguistic features that have already been captured by the pre-trained model. In this light, we propose **R**obust **I**nformative **F**ine-**T**uning (RIFT), a novel adversarial fine-tuning method from an information-theoretical perspective. In particular, RIFT encourages an objective model to retain the features learned from the pre-trained model throughout the entire fine-tuning process, whereas a conventional one only uses the pre-trained weights for initialization. Experimental results show that RIFT consistently outperforms the state-of-the-arts on two popular NLP tasks: sentiment analysis and natural language inference, under different attacks across various pre-trained language models. [1].

## 1 Introduction

Deep models are well-known to be vulnerable to adversarial examples [64, 19, 50, 35]. For instance, fine-tuned models pre-trained on very large corpora can be easily fooled by word substitution attacks using only synonyms [2, 58, 32, 12]. This has raised grand security challenges to modern NLP systems, such as spam filtering and malware detection, where pre-trained language models like BERT [11] are widely deployed.

Attack algorithms [19, 7, 76, 40, 2] aim to maliciously generate adversarial examples to fool a victim model, while adversarial defense aims at building robustness against them. Among the defense methods, adversarial training [64, 19, 48, 45] is the most effective one [3]. It updates model parameters using perturbed adversarial samples generated on-the-fly and yields consistently robust performance even against the challenging adaptive attacks [3, 68].

However, despite its effectiveness in training from scratch, adversarial training may not directly fit the current NLP paradigm, the fine-tuning of pre-trained language models. First, fine-tuning *per se* suffers from catastrophic forgetting [46, 18, 34], *i.e.*, the resultant model tends to over-fit to a small fine-tuning data set, which may deviate too far from the pre-trained knowledge [25, 81]. Second, adversarially fine-tuned models are more likely to forget: adversarial samples are usually

---

[1]Our code will be available at https://github.com/dongxinshuai/RIFT-NeurIPS2021.

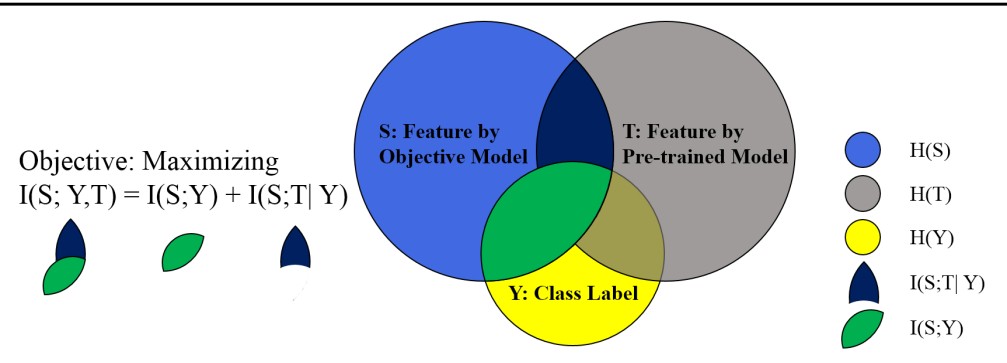

Figure 1: An illustration of the overall objective of RIFT. Maximizing $I(S;Y)$ encourages features of the objective model to be predictive of the class label, while maximizing $I(S;T|Y)$ encourages learning robust and generic linguistic information from the pre-trained model. (Random variable $S$ denotes extracted features of $X$ by the objective model and $T$ by the pre-trained language model)

out-of-distribution [38, 63], thus they are generally inconsistent with the pre-trained model. As a result, adversarial fine-tuning fails to memorize all the robust and generic linguistic features already learned during pre-training [65, 57], which are however very beneficial for a robust objective model.

Addressing forgetting is essential for achieving a more robust objective model. Conventional wisdom such as pre-trained weight decay [8, 10] and random mixout [37] mitigates forgetting by constraining $l_p$ distance between the two models' parameters. Though effective to some extent, however, it is limited because change in the model parameter space only serves as an imperfect proxy for that in the function space [5]. Besides, the extent to which an encoder fails to retain information also depends on the input distribution. Therefore, a better way to encourage memorization is favorable.

In this paper, we follow an information-theoretical lens to look into the forgetting problem: we employ mutual information to measure how well an objective model memorizes the useful features captured before. This motivates our novel adversarial fine-tuning method, Robust Informative Fine-Tuning (RIFT). In addition to fitting a down-stream task as conventions, RIFT maximizes the mutual information between the output of an objective model and that of the pre-trained model conditioned on the class label. It encourages an objective model to continuously retain useful information from the pre-trained one throughout the whole fine-tuning process, whereas the conventional one only makes use of the pre-trained weights for initialization. We illustrate the overall objective of RIFT in Figure 1 and summarize the major contributions of this paper as follows:

- To the best of our knowledge, we are the first to investigate the intensified phenomenon of catastrophic forgetting in the adversarial fine-tuning of pre-trained language models.

- To address the forgetting problem and achieve more robust models, we propose a novel approach named Robust Informative Fine-Tuning (RIFT) from an information-theoretical perspective. RIFT enables an objective model to retain robust and generic linguistic features throughout the whole fine-tuning process and thus enhance robustness against adversarial examples.

- We empirically evaluate RIFT on two prevailing NLP tasks, sentiment analysis and natural language inference, where RIFT consistently outperforms the state-of-the-arts in terms of robustness, under different attacks across various pre-trained language models.

## 2 Methodology

### 2.1 Notations and Problem Setting

In this paper, we focus on the text classification task to introduce our method, while it can be easily extended to other NLP tasks. We suppose random variables $X, Y \sim p_\mathcal{D}(x, y)$, where $X$ represents the textual input, $Y$ represents the class label, $p_\mathcal{D}$ is the data distribution, and $x, y$ are the observed values. Our goal is to build a classifier $q(y|F_s(x))$, where $F_s(\cdot)$, referred to as our objective model

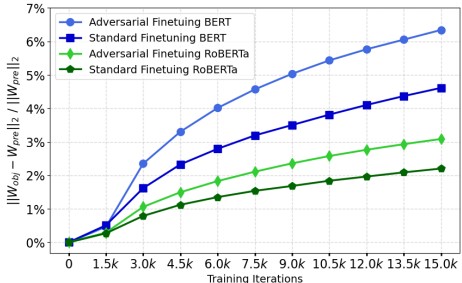 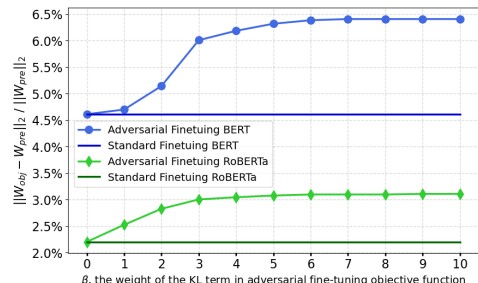

(a) In adversarial fine-tuning, the relative $L_2$ distance continuously grows as the fine-tuning proceeds.

(b) Trading clean accuracy off against robustness (by increasing $\beta$ in Eq. 2) induces increased relative $L_2$ distance at the last epoch.

Figure 2: Relative $L_2$ distance between the pre-trained model and the objective model in the parameter space, under different fine-tuning schemes on IMDB, with small learning rate 2e-5.

in the rest of this paper, is a feature extractor fine-tuned from the encoder of a pre-trained language model, *e.g.,* Transformer [72], and $q(\cdot|\cdot)$ is parameterized by using an MLP with softmaxed output. We favor a classifier that is robust against adversarial attacks [64, 2], *i.e.,* maintaining high accuracy even given adversarial examples as inputs.

## 2.2 Adversarial Fine-Tuning Suffers From Forgetting

Adversarial training [64, 19, 45] is currently the most effective defense technique [3]. Its training objective can be generally formulated as:

$$\min[\mathop{\mathbb{E}}_{x,y\sim p_{\mathcal{D}}}[\max_{\hat{x}\in\mathbb{B}(x)}\mathcal{L}(x,\hat{x},y)]], \tag{1}$$

where $\mathbb{B}(x)$ defines the allowed perturbation space around $x$, and $\mathcal{L}$ is a loss to encourage correct predictions both given vanilla samples and given perturbed examples. Such an objective function can also be applied to the fine-tuning of a pre-trained language model towards robustness, and we refer to it as adversarial fine-tuning. The loss function in Eq. 1 can be specified following Miyato et al. [48], Zhang et al. [79] in a semi-supervised fashion as:

$$-\log q(y|F_s(x)) + \beta\mathrm{KL}\big(q(\cdot|F_s(x))\|q(\cdot|F_s(\hat{x}))\big), \tag{2}$$

where the Kullback–Leibler divergence encourages invariant predictions between $x$ and $\hat{x}$.

Despite its effectiveness in training from scratch, such an adversarial objective function may not directly fit the fine-tuning scenario. To begin with, fine-tuning itself suffers from catastrophic forgetting [46, 18, 34]. During fine-tuning, an objective model $F_s(\cdot)$ tends to continuously deviate away from the pre-trained one to fit a down-stream task [25, 81], and thus the useful information already captured before is less utilized as the fine-tuning proceeds. Further, adversarial fine-tuning suffers even more from the forgetting problem, the reasons of which are given as what follows.

**(i) Adversarial Fine-Tuning Tends to Forget:** Adversarial fine-tuning targets at tackling adversarial examples, which are generally out of the manifold [38, 63] of the pre-training corpora. To additionally handle them, an objective model would be fine-tuned towards a solution that is far away from the optimization starting point, *i.e.*, the pre-trained model. Figure 2 (b) empirically shows this point: by increasing $\beta$ in Eq. 2, we emphasize more on robustness instead of vanilla accuracy, and consequently, at the last epoch the distance between models also increases. Besides, adversarial fine-tuning often entails more iterations to converge (several times of normal fine-tuning), which further intensifies the forgetting problem, as the objective model is continuously deviating away as shown in Figure 2 (a).

**(ii) Adversarial Fine-Tuning Needs to Memorize:** Overfitting to training data is a dominant phenomenon throughout the adversarial training/fine-tuning process [60]. To mitigate overfitting and generalize better, adversarial fine-tuning should retain all the generalizable features already captured by the pre-trained model [11, 41]. In addition, adversarial fine-tuning favors an objective model that

extracts features invariant to perturbations [70, 28] for robustness. As such, all those generalizable and robust linguistic information captured during pre-training [11, 65] are particularly beneficial and should be memorized. We empirically validates that encouraging memorization does improve both generalization and robustness in Sec. 3.5.

To address forgetting, previous methods such as pre-trained weight decay [8, 10] and random mixout [37] have shown their effectiveness in stabilizing fine-tuning [81]. However, they focus only on the parameter space, in which they encourage an object model to be similar to the pre-trained one. Distance in the parameter space can only approximately characterize the change in the function space, and fails to take the data distribution into consideration. A more natural way to capture the extent to which a model memorizes or forgets, should be using the mutual information between outputs of the two models, and we provide our solution as follows.

## 2.3 Informative Fine-Tuning

We first use an information-theoretical perspective to look into how a pre-trained model should be leveraged throughout the whole fine-tuning process.

We define random variable $T = F_t(X)$ as the feature of $X$ extracted by the pre-trained language model $F_t(\cdot)$, and $t$ as the observed value of $T$. Similarly, we define random variable $S = F_s(X)$ as the feature of $X$ extracted by our objective model $F_s(\cdot)$, and $s$ as the observed value of $S$. We formulate an overall fine-tuning objective as follows. The motivation is to train $F_s(\cdot)$ such that $S$ is capable of predicting $Y$, as well as preserving the information from $T$.

$$\max I(S;\, Y, T), \tag{3}$$

*i.e.,* maximizing the mutual information between (i) the feature extracted by the objective model and (ii) the class label plus the feature extracted by the pre-trained model. Since the pre-trained model $F_t(\cdot)$ is a fixed deterministic function, the fine-tuning objective in Eq. 3 in essense encourages $F_s(\cdot)$ to output features that contain as much information as possible for predicting $Y$ and $T$. This enables the objective model to learn from the pre-trained language model via $T$ throughout the whole fine-tuning process, and thus helps address the forgetting problem.

However, the objective defined in Eq. 3 is generally hard to optimize directly. Therefore, we decompose Eq. 3 into two terms as follows:

$$I(S;\, Y, T) = I(S; Y) + I(S; T \mid Y), \tag{4}$$

where $I(S; Y)$ measures how well the output of our objective model can predict the label, and $I(S; T \mid Y)$ measures when conditioned on the class label, how well the output features of the two models can predict each other. Visualization of each component can be seen in Figure 1 for more intuitive understandings. We next introduce how each term in the right-hand side of Eq. 4 can be transformed into a tractable lower bound for optimization.

**(i) Maximizing I(S;Y):** We treat $q(y|s)$, which is the classification layer that takes the features extracted by the objective model as input, as a variational distribution of $p(y|s)$, and derive a variational lower bound on $I(S; Y)$ following variational inference [33] as follows:

$$I(S; Y) = H(Y) - \mathbb{E}_{x,y \sim p_{\mathcal{D}}}\big[-\log q(y|s)\big] + \mathrm{KL}\big(p(\cdot|s)\|q(\cdot|s)\big) \tag{5}$$

$$\geq H(Y) - \mathbb{E}_{x,y \sim p_{\mathcal{D}}}\big[-\log q(y|s)\big], \tag{6}$$

where $H(Y)$ is a constant measuring the shannon entropy of $Y$, and $\mathbb{E}_{X,Y}\big[-\log q(y|s)\big]$ is essentially the cross-entropy loss using $q(y|s)$ for classification. Then, the objective of maximizing $I(S; Y)$ can be achieved by minimizing $\mathbb{E}_{X,Y}\big[-\log q(y|s)\big]$ instead.

**(ii) Maximizing I(S;T| Y):** The definition of the conditional mutual information $I(S; T \mid Y)$ is as follows:

$$I(S; T \mid Y) = \mathbb{E}_{y \sim p_{\mathcal{D}}(y)}\big[I(S; T) \mid Y = y\big] = \mathbb{E}_{y \sim p_{\mathcal{D}}(y)}\Big[\, \mathbb{E}_{x \sim p_{\mathcal{D}}(x|y)}\big[\log \frac{p(s,t|y)}{p(s|y)p(t|y)}\big]\Big]. \tag{7}$$

To achieve a tractable objective for maximizing Eq. 7, we employ noise contrastive estimation [20, 49] and derive a lower bound $-\mathcal{L}_{\mathrm{info}}$ on the conditional mutual information. This can be summarized in the following Lemma (the proof of which can be found in Appendix A.2):

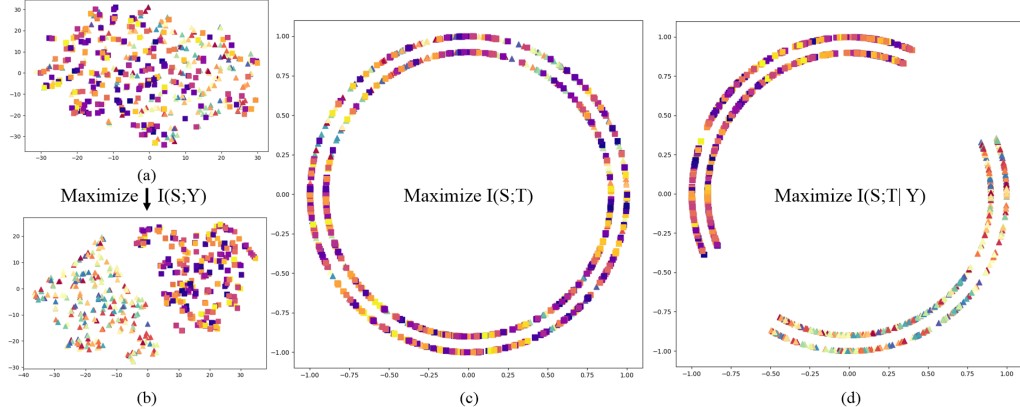

Figure 3: Visualization of the learned geometry using 500 random samples from IMDB by t-SNE [71]. ▲ and ■ denote samples from two different classes respectively. Different colors represent different data point IDs (approximately due to limited color space). In (c) and (d), $F_s(x)$ are projected to $\mathcal{S}^1$ with radius 1, while $F_t(x)$ with radius 0.9. For each sub-image: (a) The geometry of $F_s(x)$ before fine-tuning, represents the data manifold. (b) Maximizing $I(S;Y)$ encourages separating the whole data manifold into two class-specific data manifolds. (c) Maximizing $I(S;T)$ by contrastive loss in essence encourages alignment between two circles and uniformity over the whole data manifold. (d) Maximizing $I(S;T|Y)$ by contrastive loss in essence encourages alignment and uniformity inside each class-specific data manifold. Best view in color with zooming in.

**Lemma 1.** *Given* $\{x_i, y\}_{i=1}^N$ *that is sampled i.i.d. from* $p_{\mathcal{D}}(x|y)$, $s_i = F_s(x_i)$, *and* $t_i = F_t(x_i)$, $I(S;T|Y)$ *is lower bounded by* $-\mathcal{L}_{info} = \mathbb{E}_{y\sim p_{\mathcal{D}}(y)}\big[\mathbb{E}_{\{x_i,y\}_{i=1}^N}\big[\frac{1}{N}\sum_{i=1}^N \log \frac{e^{f_y(s_i,t_i)}}{\sum_{j=1}^N e^{f_y(s_i,t_j)}} + \log N\big]\big]$, *and* $f_y$ *is a score function indexed by* $y$.

By leveraging Lemma 1, $\mathcal{L}_{info}$ can be computed using a batch of samples and then minimized for maximizing $I(S;T|Y)$. The score function $f_y$ is defined as the inner product after non-linear projections into a space of hyper-sphere following [9] as $f_y(a,b) = \frac{1}{\tau}\frac{\langle g_y^1(a), g_y^2(b)\rangle}{\|g_y^1(a)\|_2 \|g_y^2(b)\|_2}$, where $g_y^1$ and $g_y^2$ are parameterized by using MLPs.

## 2.4 Conditional Mutual Information Better Fits A Down-Stream Task

Above we have introduced the training objective of informative fine-tuning and decomposed it into $I(S;Y)$ and $I(S;T|Y)$ for optimization. However, one may wonder why not maximize $I(S;T)$ directly instead of $I(S;T|Y)$. From an information-theoretical perspective, if we maximize $I(S;Y)$ and $I(S;T)$, the intersection of the three circles in Figure 1, *i.e.,* the interaction information of them, is repeatedly optimized and might induce confliction. To further elaborate this point, we look into contrastive loss and give an explanation as follows.

We consider contrastive learning by decomposing it into the encouragement of alignment and uniformity following [74]. For example, maximizing $I(S;T) = H(S) - H(S|T)$ is to decrease $H(S|T)$ and increase $H(S)$. Decreasing $H(S|T)$ corresponds to encouraging alignment in contrastive learning, which aims to align $s_i$ and $t_i$ in the sense of inner product when they are projected into a hyper-sphere. In the meanwhile, increasing $H(S)$ corresponds to uniformity in contrastive learning, which encourages $s_i$ and $s_j$, where $i \neq j$, to diffuse over the whole sub-space as uniformly as possible. It can be seen in Figure 3 (c) that, in the space of $\mathcal{S}^1$, alignment is achieved in that points from two different circles but with the same color are aligned, and uniformity is achieved in that all points diffuse over the whole $\mathcal{S}^1$.

However, when maximizing $I(S;T)$, the uniformity in the contrastive loss is encouraged over the whole data manifold. Diffusing uniformly over the whole data manifold can be against the objective of maximizing $I(S;Y)$, which aims to separate the whole data manifold into class-specific parts as shown in Figure 3 (b). In contrast, maximizing $I(S;T|Y)$ only encourages uniformity inside each class-specific data manifold, which is complementary to maximizing $I(S;Y)$, as show in Figure 3 (d). In the meanwhile, the alignment is still enforced. More empirical support can be found in Sec. 3.4.

**Algorithm 1** RIFT

---

**Input**: dataset $\mathcal{D}$, hyper-parameters of AdamW [43]
**Output**: the model parameters $\theta$ and $\phi$

 1: Initialize $\theta$ using the pre-trained model, and initialize $\phi$ and $\varphi$ randomly.
 2: **repeat**
 3:      Sample $y \sim p_{\mathcal{D}}(y)$ and then $\{x_i, y\}_{i=1}^N \sim p_{\mathcal{D}}(x|y)$
 4:      **for** every $x_i, y$ in the mini-batch $\{x_i, y\}_{i=1}^N$ **do**
 5:          Find $\hat{x}_i$ by solving Eq. 8;
 6:      **end for**
 7:      Compute the loss function defined in Eq. 11 and update $\theta$, $\phi$, and $\varphi$ by gradients.
 8: **until** the training converges.

---

## 2.5 Robust Informative Fine-Tuning

In this section, we are going to put the objective function of informative fine-tuning into the adversarial context and introduce Robust Informative Fine-Tuning (RIFT).

To robustly train a model, the adversarial examples $\hat{x}$ for training should be formulated and generated first. As our end goal is to enhance robustness in down-stream tasks, the generation process of adversarial examples should focus on preventing a model from predicting the ground truth label. However, using label for generating adversarial examples in training often induces label leaking [36], *i.e.,* the generated adversarial examples contains label information which is used as a trivial short-cut by a classifier. As such, we follow [48, 79] to generate $\hat{x}$ in a self-supervised fashion:

$$\hat{x} = \underset{x' \in \mathbb{B}(x)}{\arg\max} \ \mathrm{KL}\big(q(\cdot|F_s(x))\|q(\cdot|F_s(x'))\big). \tag{8}$$

By solving Eq. 8, $\hat{x}$ is found to induce the most different prediction from that of a vanilla sample $x$ in terms of KL, inside the attack space $\mathbb{B}(x)$.

Next, we introduce how to robustly optimize each objective in the informative fine-tuning by using $\hat{x}$.

**(i) Robustly Maximizing I(S;Y):** As shown in Sec. 2.3, maximizing $I(S;Y)$ can be achieved by minimizing a cross-entropy loss instead. To encourage adversarial robustness, this cross-entropy loss can be upgraded to encourage both correct predictions on and invariant predictions between $x$ and $\hat{x}$ [48, 79]. We formulate such an objective function as follows:

$$\min_{\theta,\phi} \mathcal{L}_{\text{r-task}}, \quad \mathcal{L}_{\text{r-task}} = \underset{x,y \sim p_{\mathcal{D}}}{\mathbb{E}} \big[ -\log q(y|F_s(x)) + \beta \mathrm{KL}\big(q(\cdot|F_s(x))\|q(\cdot|F_s(\hat{x}))\big) \big], \tag{9}$$

where $\theta$ denotes the parameters of $F_s(\cdot)$ and $\phi$ denotes the parameters of $q(\cdot|\cdot)$. Minimizing $\mathcal{L}_{\text{r-task}}$ in Eq. 9 corresponds to adversarilly maximizing $I(S;Y)$ for robust performance in a down-stream task. By doing so, the adversarial example $\hat{x}$ is generated by Eq. 8 first and then both $x$ and $\hat{x}$ are used to optimize the model parameters $\theta$ and $\phi$.

**(ii) Robustly Maximizing I(S;T| Y):** We aim to maximize the conditional mutual information $I(S;T| Y)$, but under an adversarial distribution of input data. We formulate such a term as $I(\hat{S};T| Y)$, where random variable $\hat{S} = F_s(\hat{X})$, $\hat{X} \sim p_{\text{adv}}(\hat{x}|x, \theta, \phi, \mathbb{B})$, and sampling from $p_{\text{adv}}$, the adversarial distribution of input data, is to generate adversarial example $\hat{x}$ by Eq. 8.

To optimize $I(\hat{S};T| Y)$, we propose $-\mathcal{L}_{\text{r-info}}$ as a lower bound on it, and formulate the objective to minimize $\mathcal{L}_{\text{r-info}}$ as follows (similar to $-\mathcal{L}_{\text{info}}$ by using Lemma 1):

$$\min_{\theta,\varphi} \mathcal{L}_{\text{r-info}}, \quad \mathcal{L}_{\text{r-info}} = \underset{y \sim p_{\mathcal{D}}(y)}{\mathbb{E}} \Big[ \underset{\{x_i,y\}_{i=1}^N \sim p_{\mathcal{D}}(x|y)}{\mathbb{E}} \big[ \frac{1}{N} \sum_{i=1}^N - \log \frac{e^{f_y(\hat{s}_i, t_i)}}{\sum_{j=1}^N e^{f_y(\hat{s}_i, t_j)}} - \log N \big] \Big], \tag{10}$$

where $\hat{s}_i = F_s(\hat{x}_i)$, $t_i = F_t(x_i)$, and $\varphi$ denotes the parameters of all the score functions $f_y$. By Eq. 10, we are able to encourage $F_s(\cdot)$ to retain information from $F_t(\cdot)$ in a robust fasion.

Noted that, in $\mathcal{L}_{\text{r-info}}$ we do not use $\hat{x}_i$ to extract features from the pre-trained model. This follows the spirit of Knowledge Distillation [22] (though the setting is different in that a student is expected to perform identically to a teacher in knowledge distillation, while in fine-tuning it does not): the data used to extract features of a teacher should be inside the domain for which the teacher is trained. A

Table 1: Accuracy(%) of different fine-tuning methods under attacks on IMDB.

| Method | Model | Genetic | PWWS | Method | Model | Genetic | PWWS |
|---|---|---|---|---|---|---|---|
| Standard | BERT | $38.1_{\pm2.5}$ | $40.7_{\pm1.1}$ | Standard | RoBERTa | $42.1_{\pm2.1}$ | $45.6_{\pm3.1}$ |
| Adv-Base | BERT | $74.8_{\pm0.4}$ | $68.3_{\pm0.3}$ | Adv-Base | RoBERTa | $70.3_{\pm1.2}$ | $63.3_{\pm0.7}$ |
| Adv-PTWD | BERT | $73.9_{\pm0.4}$ | $69.1_{\pm0.7}$ | Adv-PTWD | RoBERTa | $69.3_{\pm1.4}$ | $64.4_{\pm0.3}$ |
| Adv-Mixout | BERT | $75.4_{\pm0.7}$ | $68.8_{\pm0.6}$ | Adv-Mixout | RoBERTa | $70.6_{\pm1.0}$ | $63.9_{\pm1.3}$ |
| **RIFT** | BERT | $\mathbf{77.2}_{\pm0.8}$ | $\mathbf{70.1}_{\pm0.5}$ | **RIFT** | RoBERTa | $\mathbf{73.5}_{\pm0.8}$ | $\mathbf{66.3}_{\pm0.7}$ |

(a) Accuracy (%) based on BERT-base-uncased.     (b) Accuracy (%) based on RoBERTa-base.

Table 2: Accuracy(%) of different fine-tuning methods under attacks on SNLI.

| Method | Model | Genetic | PWWS | Method | Model | Genetic | PWWS |
|---|---|---|---|---|---|---|---|
| Standard | BERT | $40.1_{\pm0.7}$ | $19.4_{\pm0.4}$ | Standard | RoBERTa | $43.4_{\pm1.2}$ | $20.4_{\pm1.0}$ |
| Adv-Base | BERT | $75.7_{\pm0.5}$ | $72.9_{\pm0.2}$ | Adv-Base | RoBERTa | $82.6_{\pm0.6}$ | $79.9_{\pm0.7}$ |
| Adv-PTWD | BERT | $75.2_{\pm1.0}$ | $72.6_{\pm0.5}$ | Adv-PTWD | RoBERTa | $81.2_{\pm0.8}$ | $78.9_{\pm0.7}$ |
| Adv-Mixout | BERT | $76.3_{\pm0.8}$ | $73.2_{\pm1.0}$ | Adv-Mixout | RoBERTa | $82.6_{\pm0.9}$ | $80.6_{\pm0.3}$ |
| **RIFT** | BERT | $\mathbf{77.5}_{\pm0.9}$ | $\mathbf{74.3}_{\pm1.1}$ | **RIFT** | RoBERTa | $\mathbf{83.5}_{\pm0.8}$ | $\mathbf{81.1}_{\pm0.4}$ |

(a) Accuracy (%) based on BERT-base-uncased.     (b) Accuracy (%) based on RoBERTa-base.

down-stream NLP task, *e.g.,* sentiment analysis, has a data domain that is generally a sub-domain of the pre-training corpora, while $\hat{x}_i$ can be significantly off the pre-training data manifold.

**Objective Function of Robust Informative Fine-Tuning (RIFT):** The objective function of RIFT is a combination of $\mathcal{L}_{\text{r-task}}$ and $\mathcal{L}_{\text{r-info}}$, defined as follows:

$$\min_{\theta,\phi,\varphi} \mathcal{L}_{\text{r-task}} + \alpha\mathcal{L}_{\text{r-info}}, \tag{11}$$

where the hyper-parameter $\alpha$ controls to what extent we encourage an objective model to absorb information from the pre-trained one (ablation on $\alpha$ can be seen in Sec. 3.5). We summarize the whole training process in Algorithm 1 and include more implementation details in Appendix A.1.

# 3 Experiments

## 3.1 Experimental Setting

**Tasks and Datasets:** We evaluate the robust accuracy and compare our method with the state-of-the-arts on: (i) Sentiment analysis using the IMDB dataset [44]. (ii) Natural language inference using the SNLI dataset [6]. We mainly focus on robustness against adversarial word substitutions [2, 58], as such attacks preserve the syntactic and semantics very well [31, 78, 12], and are very hard to detect even by humans. Under this attack setting, any word in the input sequence can be substituted by a semantically similar word of it (often its synonym). We evaluate robustness on 1000 random examples from the testset of IMDB and SNLI respectively following [31, 12].

**Model Architectures:** We examine our methods and compare with state-of-the-arts on the following two prevailing pre-trained language models: (i) BERT-base-uncased [11]. (ii) RoBERTa-base [41].

**Attack Algorithms:** Two powerful attacks are employed: (i) Genetic [2] based on population algorithm. Aligned with [31, 12], the population size and iterations are set as 60 and 40 respectively. (ii) PWWS [58] based on word saliency. We only attack hypothesis on SNLI aligned with [31, 12].

**Substitution Set:** We follow [31, 12] to use the substitution set from [2], and the same language model constraint is applied to Genetic attacks and not to PWWS attacks.

## 3.2 Comparative Methods

**(i) Standard Fine-Tuning:** The standard fine-tuning process first initializes the objective model by the pre-trained weight, and then use cross-entropy loss to fine-tune the whole model.

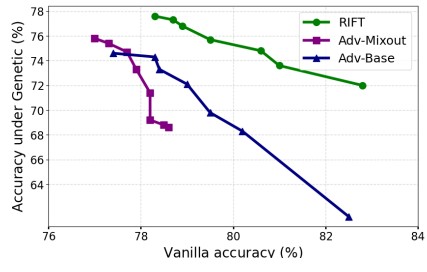

(a) Accuracy (%) under Genetic attacks.

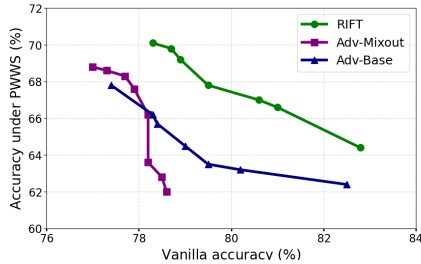

(b) Accuracy (%) under PWWS attacks.

Figure 4: Tradeoff curve between robustness and vanilla accuracy of BERT-based model on IMDB.

Table 3: Accuracy(%) of RIFT with maximizing $I(S;T|Y)$ and $I(S;T)$ respectively.

| Maximizing | Model | Genetic | PWWS | Maximizing | Model | Genetic | PWWS |
|---|---|---|---|---|---|---|---|
| $\mathbf{I(S;T|Y)}$ | BERT | **77.2** | **70.1** | $\mathbf{I(S;T|Y)}$ | BERT | **77.5** | **74.3** |
| $I(S;T)$ | BERT | 76.1 | 69.4 | $I(S;T)$ | BERT | 76.6 | 72.1 |
| $\mathbf{I(S;T|Y)}$ | RoBERTa | **73.5** | **66.3** | $\mathbf{I(S;T|Y)}$ | RoBERTa | **83.5** | **81.1** |
| $I(S;T)$ | RoBERTa | 72.0 | 65.3 | $I(S;T)$ | RoBERTa | 82.5 | 79.4 |

(a) Accuracy (%) under attacks on IMDB.        (b) Accuracy (%) under attacks on SNLI.

**(ii) Adversarial Fine-Tuning Baseline (Adv-Base):** We employ the state-of-the-art defense against word substitutions, ASCC-Defense [12], as the adversarial fine-tuning baseline. This method is not initially proposed for pre-trained language models but can readily extend to perform adversarial fine-tuning. During fine-tuning, adversarial example $\hat{x}$, which is a sequence of convex combinations, is generated first and then both $x$ and $\hat{x}$ are used for optimization using objective defined in Eq. 9.

**(iii) Adv + Pre-Trained Weight Decay (Adv-PTWD):** Pre-trained weight decay [8, 10] penalizes $\lambda\|W_{\text{obj}} - W_{\text{pre}}\|_2$, and mitigates catastrophic forgetting [75, 37]. We combine it with the adversarial baseline for comparisons and $\lambda$ is chosen as 0.01 on IMDB and 0.005 on SNLI for best robustness.

**(iv) Adv + Mixout (Adv-Mixout):** Motivated by Dropout [62] and DropConnect [73], Mixout [37] is proposed to addresses catastrophic forgetting in fine-tuning. At each iteration each parameter is replaced by its pre-trained counter-part with probability $m$. We combine it with the adversarial fine-tuning baseline for comparisons with our method and $m$ is chosen as 0.6 for best robustness.

**(v) Robust Informative Fine-Tuning (RIFT):** The proposed adversarial fine-tuning method. It can be deemed as the adversarial fine-tuning baseline plus the $\mathcal{L}_{\text{r-info}}$ term. We set $\tau$ as 0.2 for all score functions $f_y$. For best robust accuracy, $\alpha$ is chosen as 0.1 and 0.7 on IMDB and SNLI respectively. Ablation study on $\alpha$ can be seen in Sec. 3.5.

For fair comparisons, all compared adversarial fine-tuning methods use the same $\beta$ on a same dataset, *i.e.*, $\beta = 10$ on IMDB and $\beta = 5$ on SNLI, both of which are chosen for the best robust accuracy. Early stopping [60] is used for all compared methods according to best robust accuracy More implementation details and runtime analysis can be found in Appendix A.1 and A.3.

### 3.3 Main Result

In this section we compare our method with state-of-the-arts by robustness under attacks. As shown in Tables 1 and 2., RIFT consistently achieves the best robust performance among state-of-the-arts in all datasets across different pre-trained language models under all attacks. For instance, on IMDB our method outperforms the RoBERTa-based runner-up method by 2.9% under Genetic attacks and 1.9% under PWWS attacks. On SNLI based on BERT, we surpass the runner-up method by 1.2% under Genetic attacks and 1.1% under PWWS attacks. In addition, RIFT consistently improves robustness upon the adversarial fine-tuning baseline, while the improvements by other methods are not stable. We observe that on SNLI, we surpass the runner-up by a relatively small margin compared to IMDB. This may relate to the dataset property: input from SNLI has a smaller attack space (on average $6.5^4$

Table 4: Ablation study on the hyper-parameter $\alpha$.

| Parameter $\alpha$ | Vanilla | Genetic | PWWS | Parameter $\alpha$ | Vanilla | Genetic | PWWS |
|---|---|---|---|---|---|---|---|
| 0.00 | 78.1 | 74.8 | 68.3 | 0.00 | 79.4 | 75.7 | 72.9 |
| 0.05 | **78.4** | 76.5 | 69.2 | 0.50 | 80.0 | 76.9 | 73.8 |
| **0.10** | 78.3 | **77.2** | **70.1** | **0.70** | 80.4 | **77.5** | **74.3** |
| 0.30 | 78.3 | 76.2 | 69.5 | 1.00 | **80.6** | 77.3 | 73.2 |

(a) Acc (%) of RIFT based on BERT on IMDB.  (b) Acc (%) of RIFT based on BERT on SNLI.

combinations of substitutions on SNLI compared to $6^{108}$ on IMDB), and thus smaller absolute space left for improvement. For results of vanilla accuracy please refer to Appendix A.4.

## 3.4 Conditional Mutual Information Does Fit a Down-Stream Task Better

In this section we empirically validate that maximizing $I(S;T|Y)$ cooperates with a down-stream task better than maximizing $I(S;T)$ does. We plot two sets of results of our method with maximizing $I(S;T|Y)$ and $I(S;T)$ respectively (using similar noise contrastive loss with all other hyper-parameters the same). As shown in Table 3, RIFT with maximizing $I(S;T|Y)$ consistently outperforms that with maximizing $I(S;T)$ in terms of robustness under all attacks on both IMDB and SNLI, which serves as an empirical validation of Sec. 2.4.

## 3.5 Ablation Study and Tradeoff Curve Between Robustness and Vanilla Accuracy

In this section we conduct ablation study on $\alpha$, which is the weight of $\mathcal{L}_{\text{r-info}}$ in Eq. 11. It aims to control to what extent the objective model absorbs information from the pre-trained one. As shown in Table 4, a good value of $\alpha$ improves both vanilla accuracy and robust accuracy; *e.g.*, on IMDB increasing $\alpha$ from 0 to 0.1 results in increased vanilla accuracy by $0.2\%$ and increased robust accuracy under Genetic attacks by $2.4\%$. This demonstrates that RIFT does motivate the objective model to retain robust and generalizable features that are beneficial to both robustness and vanilla accuracy. When $\alpha$ goes too large, the objective of fine-tuning would focus too much on preserving as much information from the pre-trained model as possible, and thus ignores the down-stream task.

One may wonder whether informative fine-tuning itself can improve the clean accuracy upon normal fine-tuning. The answer is yes; *e.g.*, on IMDB using RoBERTa, informative fine-tuning improves about $0.3\%$ clean accuracy upon normal fine-tuning baseline. The improvement is not very significant as normal fine-tuning targets at vanilla input only and thus suffers less from the forgetting problem. One common problem in contrastive loss is that the complexity grows quadratically with $N$, but larger $N$ contributes to tighter bound on the mutual information [49, 53]. One potential improvement is to maintain a dictionary updated on-the-fly like [21] and we will leave it for future exploration.

We finally show the trade-off curve between robustness and vanilla accuracy in Figure 4. As shown, RIFT outperforms the state-of-the-arts in terms of both robust accuracy and vanilla accuracy. It again validates that RIFT indeed helps the objective model capture robust and generalizable information to improve both robustness and vanilla accuracy, rather than trivially trading off one against another.

## 4 Related Work

**Pre-Trained Language Models:** Language modeling aims at estimating the probabilistic density of the textual data distribution. Classical language modeling methods vary from CBOW, Skip-Gram [47], to GloVe [51] and ELMo [52]. Recently, Transformer [72] has demonstrated its power in language modeling, *e.g.,* GPT [55] and BERT [11]. The representations learned by deep pre-trained language models are shown to be universal and highly generalizable [77] [41], and fine-tuning pre-trained language models becomes a paradigm in many NLP tasks [25, 11].

**Adversarial Robustness:** DNNs have achieved success in many fields, but they are susceptible to adversarial examples [64]. In NLP, attacks algorithms include char-level modifications [24, 13, 4, 17, 15, 54], sequence-level manipulations [29, 59, 30, 82], and adversarial word substitutions [2, 40, 58, 32, 80, 78]. For defenses, adversarial training [64, 19, 45] generates adversarial examples on-the-fly and is currently the most effective. Miyato et al. [48] first introduce adversarial training to

NLP using $L_2$-ball to model perturbations, and later more geometry-aligned methods like first-order approximation[13], axis-aligned bound [31, 27], and convex hull [12], become favorable.

**Fine-Tuning and Catastrophic Forgetting:** The fine-tuning of pre-trained language models [25, 11] can be very unstable [11], as during fine-tuning the objective model can deviate too much from the pre-trained one, and easily over-fits to small fine-tuning sets [25]. Such phenomenon is referred as catastrophic forgetting [46, 18] in fine-tuning [25, 37, 81]. Methods to address catastrophic forgetting include pre-trained weight decay [8, 10, 81], learning rate decreasing [25], and Mixout regularization [37]. These methods focus on parameter space to constrain the distance between two models while our method addresses forgetting following an information-theoretical perspective. In continuous learning, method like rehearsal-based [42, 56, 39, 1] and regularization-based [34, 14, 26] also aims at the forgetting problem, but under a different setting: they focus on balanced performance on tasks with previous data, while in our setting the pre-training corpora are not available and language modeling is not our concern.

**Contrastive Learning and Mutual Information:** Contrastive learning [49, 23, 61] has demonstrated its power in learning self-supervised representations [9, 21]. The contrastive loss such as InfoNCE[49], InfoMax [23], is proposed as maximizing the mutual information of features in different views, and further discussed in [53, 67, 69, 66, 74]. This paper shares the same perspective of information theory, but looks into a different problem, addressing forgetting for fine-tuning. Addressing forgetting motivates our objective of maximizing the mutual information between the outputs of two models, which is different from conventional contrastive learning, and cooperation with fine-tuning motivates the use of conditional mutual information, which is quite different also. Fischer and Alemi [16] propose to restrict information for robustness in vision domain. Restricting information to some extent helps a model ignore spurious features, but may be at the cost of ignoring robust features as well. In contrast, the proposed method aims to retain the generic and robust features that are already captured by a pre-trained language model, and thus better fits a fine-tuning scenario.

## 5  Discussion and Conclusion

In this paper, we proposed RIFT to fine-tune a pre-trained language model towards robust down-stream performance. RIFT addresses the forgetting problem from an information-theoretical perspective and consistently outperforms state-of-the-arts in our experiments. We hope that this work can contribute to the NLP robustness in general and thus to more reliable NLP systems in real-life applications.

## Acknowledgements

This work is supported by the Singapore Ministry of Education (MOE) Academic Research Fund (AcRF) Tier 1 (S21/20) and Tier 2.

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
