# Supplementary Material of
# How Should Pre-Trained Language Models Be Fine-tuned Towards Adversarial Robustness?

**Xinshuai Dong**
Nanyang Technological University & Sea AI Lab
dongxinshuai@outlook.com

**Luu Anh Tuan**
Nanyang Technological University
anhtuan.luu@ntu.edu.sg

**Min Lin**
Sea AI Lab
linmin@sea.com

**Shuicheng Yan**
Sea AI Lab
yansc@sea.com

**Hanwang Zhang**
Nanyang Technological University
hanwangzhang@ntu.edu.sg

## A   Appendix

### A.1   Implementation Details

**Textual sequence processing.** For consistent word numbers per input under word substitution attacks, we seperate word-level tokens by space and punctuations, and then follow the original tokenizer of BERT/RoBERTa to tokenize the input sequence. The byte-level RoBERTa tokenizer is further modified to output one token per word to fit the setting of word substitution attacks. The maximum number of tokens including special tokens per input is set as 300 for IMDB, and 80 for SNLI.

**Hyper-parameters and optimization details.** We set $\alpha$ as 0.1 for IMDB and 0.7 for SNLI. $\tau$ is set as 0.2 for both IMDB and SNLI. Other hyper-parameters are set as the same among all compared methods for fair comparisons. For both standard fine-tuning and adversarial fine-tuning, we run for 20 epochs with batch size 32 for IMDB, and run for 20 epochs with batch size 120 for SNLI. Early stopping is used for all compared methods according to best robust accuracy. AdamW optimizer is employed with learning rate of 0.00002. We do not apply weight decay on an objective model, and set weight decay rate as 0.0002 for task-specific layers.

**Model architectures.** For both BERT and RoBERTa, the representation with respect to the sequence classification token of the last layer is employed as the output feature, which is later taken as the input of the task-specific layers for predictions. The task-specific layer is a MLP that has two linear layers with relu activation after the first layer and softmax after the second one.

### A.2   The Proof of Lemma 1

The loss $\mathcal{L}_{\text{info}}$ is the categorical cross-entropy loss of identifying $t_i$ among $\{t_j\}_{j=1}^N$, given $s_i$ and $y$. Thus, the optimal $e^{f_y(s,t)}$ that minimizes $\mathcal{L}_{\text{info}}$ is proportional to $\frac{p(t|s,y)}{p(t|y)}$ (refer to [6] for more

35th Conference on Neural Information Processing Systems (NeurIPS 2021).

Table 5: Vanilla Accuracy(%) of different fine-tuning methods on IMDB.

| Method | Model | Vanilla Accuracy | Method | Model | Vanilla Accuracy |
|--------|-------|------------------|--------|-------|------------------|
| Standard | BERT | 93.1 | Standard | RoBERTa | 94.9 |
| Adv-Base | BERT | 74.6 | Adv-Base | RoBERTa | 80.1 |
| Adv-PTWD | BERT | 76.6 | Adv-PTWD | RoBERTa | 80.7 |
| Adv-Mixout | BERT | 77.8 | Adv-Mixout | RoBERTa | 79.0 |
| **RIFT** | BERT | **78.3** | **RIFT** | RoBERTa | **84.2** |

(a) Accuracy (%) based on BERT-base-uncased.     (b) Accuracy (%) based on RoBERTa-base.

Table 6: Vanilla Accuracy(%) of different fine-tuning methods on SNLI.

| Method | Model | Vanilla Accuracy | Method | Model | Vanilla Accuracy |
|--------|-------|------------------|--------|-------|------------------|
| Standard | BERT | 89.2 | Standard | RoBERTa | 91.3 |
| Adv-Base | BERT | 79.4 | Adv-Base | RoBERTa | 87.1 |
| Adv-PTWD | BERT | 78.4 | Adv-PTWD | RoBERTa | 85.9 |
| Adv-Mixout | BERT | 79.3 | Adv-Mixout | RoBERTa | 87.1 |
| **RIFT** | BERT | **80.5** | **RIFT** | RoBERTa | **87.9** |

(a) Accuracy (%) based on BERT-base-uncased.     (b) Accuracy (%) based on RoBERTa-base.

details). We then insert $\frac{p(t|s,y)}{p(t|y)}$ into $\mathcal{L}_{\text{info}}$ and get what follows:

$$\mathcal{L}_{\text{info}} = \mathop{\mathbb{E}}_{y \sim p_{\mathcal{D}}(y)} \Big[ \mathop{\mathbb{E}}_{\{x_i,y\}_{i=1}^N} \Big[ \frac{1}{N} \sum_{i=1}^N - \log N \frac{\frac{p(t_i|s_i,y)}{p(t_i|y)}}{\sum_{j=1}^N \frac{p(t_j|s_i,y)}{p(t_j|y)}} \Big] \Big] \tag{12}$$

$$= \mathop{\mathbb{E}}_{y \sim p_{\mathcal{D}}(y)} \Big[ \mathop{\mathbb{E}}_{\{x_i,y\}_{i=1}^N} \Big[ \frac{1}{N} \sum_{i=1}^N \log \frac{1}{N} \frac{\frac{p(t_i|s_i,y)}{p(t_i|y)} + \sum_{j\neq i}^N \frac{p(t_j|s_i,y)}{p(t_j|y)}}{\frac{p(t_i|s_i,y)}{p(t_i|y)}} \Big] \Big] \tag{13}$$

$$= \mathop{\mathbb{E}}_{y \sim p_{\mathcal{D}}(y)} \Big[ \mathop{\mathbb{E}}_{\{x_i,y\}_{i=1}^N} \Big[ \frac{1}{N} \sum_{i=1}^N \log \frac{1}{N} \big( 1 + \frac{p(t_i|y)}{p(t_i|s_i,y)} \sum_{j\neq i}^N \frac{p(t_j|s_i,y)}{p(t_j|y)} \big) \Big] \Big] \tag{14}$$

$$= \mathop{\mathbb{E}}_{y \sim p_{\mathcal{D}}(y)} \Big[ \mathop{\mathbb{E}}_{\{x_i,y\}_{i=1}^N} \Big[ \frac{1}{N} \sum_{i=1}^N \log \big( \frac{1}{N} + \frac{N-1}{N} \mathop{\mathbb{E}}_{x \sim p_{\mathcal{D}}(x|y)} \big[ \frac{p(t_j|s_i,y)}{p(t_j|y)} \big] \frac{p(t_i|y)}{p(t_i|s_i,y)} \big) \Big] \Big] \tag{15}$$

$$= \mathop{\mathbb{E}}_{y \sim p_{\mathcal{D}}(y)} \Big[ \mathop{\mathbb{E}}_{\{x_i,y\}_{i=1}^N} \Big[ \frac{1}{N} \sum_{i=1}^N \log \big( \frac{1}{N} + \frac{N-1}{N} \frac{p(t_i|y)}{p(t_i|s_i,y)} \big) \Big] \Big] \tag{16}$$

$$\geq \mathop{\mathbb{E}}_{y \sim p_{\mathcal{D}}(y)} \Big[ \mathop{\mathbb{E}}_{\{x_i,y\}_{i=1}^N} \Big[ \frac{1}{N} \sum_{i=1}^N \log \frac{p(t_i|y)}{p(t_i|s_i,y)} \Big] \Big] \tag{17}$$

$$= \mathop{\mathbb{E}}_{y \sim p_{\mathcal{D}}(y)} \Big[ \mathop{\mathbb{E}}_{x \sim p_{\mathcal{D}}(x|y)} \big[ - \log \frac{p(t|s,y)}{p(t|y)} \big] \Big] \tag{18}$$

$$= -I(S;T \mid Y). \tag{19}$$

Eq. 16 to Eq. 17 is by Jensen's inequality. As such, $-\mathcal{L}_{\text{info}}$ is a lower bound on $I(S;T \mid Y)$ and a larger $N$ makes the bound tighter. The specific design of the score function $f_y$ does not impact the correctness of Lemma 1: when $-\mathcal{L}_{\text{info}}$ is maximized, $-\mathcal{L}_{\text{info}}$ is still a lower bound on the mutual information term. However, if the capacity of $f_y$ is limited, the bound might be loose.

### A.3 Runtime Analysis

All models are trained using the Nvidia A100 GPU and our implementation is based on PyTorch. As for IMDB, it takes about 10 GPU hours to train a BERT or RoBERTa based model using RIFT. As for SNLI, it takes about 40 GPU hours to train a BERT or RoBERTa based model using RIFT.

### A.4 Vanilla Accuracy

we here show the vanilla accuracy of each methods in Tabs. 5 and 6 as a supplement. As we can see, RIFT surpasses all other adversarial fine-tuning method in terms of vanilla accuracy. It again validates that RIFT does help retain the generalizable information learned before.

### A.5 License of Used Assets

The assets and the corresponding licenses are as follows. IMDB dataset [5]: Non-Commercial Licensing. SNLI dataset [2]: Creative Commons Attribution-ShareAlike 4.0 International License. Genetic attack [1]: MIT License. PWWS attack [7]: MIT License. Certified robustness [4]: MIT License. ASCC-defense [3]: MIT License.