# OpenReview forum: "How Should Pre-Trained Language Models Be Fine-Tuned Towards Adversarial Robustness?"
_NeurIPS.cc/2021/Conference — NeurIPS 2021 Poster_

### Official Review · Reviewer_8Qx5 · 2021-07-12

**Rating:** 7
**Confidence:** 4

**Summary:**

The paper studies the problem of adversarial fine-tuning of pretrained NLP models, that is fine-tuning them in a way that makes them robust to adversarial attacks. It proposes a new method called RIFT. RIFT alters the usual adversarial training by adding a regularization term that maximizes the conditional mutual information between feature vectors produced by the original pretrained model and the currently fine-tuned one. The motivation is to alleviate catastrophic forgetting. The experiments show RIFT getting a slightly higher adversarial robustness than the baselines methods, such as e.g. using the L2 penalty instead of the proposed regularizer.


**Limitations And Societal Impact:**

Within the scope of improving adversarial robustness of NLP models, the only limitation of the paper is that the improvements, while being significant, are not very big. The scope of the evaluation (focusing on NLP only) can be considered another limitation.

The societal impact of having more robust models should be rather positive.


**Main Review:**

Originality: the method draws inspiration from the recent unsupervised representation learning with contrastive losses. The applications of these methods to regularizing adversarial training appears to be novel.

Quality: the experimental part of the paper appears to be technically sound. There are ablation experiments that confirm the significance of the key aspects of the method. Same probably holds for proofs, but some steps were skipped, see my comments on clarity.

Clarity: the paper is mostly clearly written. Here are some questions/concerns:
- The score function $f_y$ and the MLPs $g^1_y(a)$ and $g^2_y(a)$ are not explained very clearly. How are they trained? What is their impact on the correctness of Lemma 1? The proof in Appendix 1 just skips this part.
- Some choices of terms are somewhat confusing. I’d recommend “downstream model” instead of “objective model”, “the feature vector” instead of “the feature”, “state-of-the-art methods” instead of “state-of-the-arts”
- The discussion in Section 2.4 should be compressed and be more to the point. Phrases like “Such a view provides some intuitions, but it is still vague in the sense of what is actually going on.” are not helpful in scientific writing.
- To put the reported results in context, it would be helpful to show the “vanilla accuracy” without adversarial training for all the employed dataset. For example, for SNLI I found a BERT baseline of 90%, where Table 6a in Section A.4 reports ~80%.

Significance: I find the results to have a non-negligible but limited significance. The adversarial robustness improves 2-3% from around 70-80%. That appears to still come at the cost of a significant decrease of vanilla accuracy (unclear how big though - see my clarity comments). But the baseline is even worse.

Other comments:
The RIFT regularizer is proposed with continual learning motivations in mind. It would be very interesting how a regularizer like that performs on continual learning benchmarks.
It would be highly interesting to see the impact of the batch size on the performance. The batch sizes that the paper uses are surprisingly small, especially given that the contrastive loss of RIFT appears to need a large batch size.


**Time Spent Reviewing:**

3.5

---

> ### Author Response · Authors · 2021-08-10
> **Author Response**
>
> We thank the reviewer for the insightful review and valuable feedback.
>
> Regarding the score function, the score function $f_y$ (which consists of $g_{y}^{1}$ and $g_{y}^{2}$) are parameterized by $\varphi$ (line 198) and optimized jointly with all other model parameters as in Eq.11. The specific design of $f_y$ does not impact the correctness of Lemma 1: when –$L_{info}$ in Lemma 1 is maximized, –$L_{info}$ is still a lower bound on the mutual information term (following the proof technique used in [1]). However, if the capacity of $f_y$ is limited, the bound would be loose. In practice, following [2], we tried naive cos similarity and cos similarity with MLPs projections as in line 145 and found that the latter works better, which is consistent with the above analysis. We thank the reviewer for pointing it out and we will add this discussion to the revision.
>
> Regarding the vanilla accuracy of the standard fine-tuning baseline we used, we provide the result as follows: \
> IMDB BERT-base-uncased: 93.1%. \
> IMDB RoBERTa-base: 94.9%. \
> SNLI BERT-based-uncased: 89.2%. \
> SNLI RoBERTa-base: 91.3%.
>
> We thank the reviewer and will add the above results to Appendix Table 6 as suggested.
> As pointed out by the reviewer, adversarial fine-tuning achieves better robust accuracy compared to standard fine-tuning, but at the cost of vanilla accuracy. This is because of the widely concerned trade-off phenomenon between standard accuracy and robust accuracy by the adversarial robustness community [3, 4]. In this sense, a good defense method is often expected to provide a better trade-off between vanilla accuracy and robust accuracy, instead of simply trading off vanilla accuracy for robustness. Following this spirit, we show in our experiments that the proposed method does improve both vanilla accuracy (Table 4) and robust accuracy (Tables 1 and 2), and provides a better trade-off between them as well (Figure 4).
>
> Regarding the batch size, a larger batch size would theoretically contribute to a tighter bound on the mutual information. In experiments, we found that a larger batch size does help achieve better performance. However, as adversarial training needs much more GPU memory than normal training, we can only set the batch size for our contrastive loss as big as possible. A larger batch size than we used in our experiments may further improve the performance of the proposed method. One potential way to bypass the hardware limit is to maintain a dictionary updated on the fly like [5] and we plan to leave it for future exploration.
> Regarding the suggestions on writing, we thank the reviewer and will revise accordingly.
>
> \
> [1] Aaron van den Oord, Yazhe Li, and Oriol Vinyals. Representation learning with contrastive predictive coding. arXiv preprint arXiv:1807.03748, 2018.
>
> [2] Ting Chen, Simon Kornblith, Mohammad Norouzi, and Geoffrey Hinton. A simple framework for contrastive learning of visual representations. In ICML, 2020.
>
> [3] Tsipras, Dimitris, Shibani Santurkar, Logan Engstrom, Alexander Turner, and Aleksander Madry. Robustness may be at odds with accuracy. In ICLR, 2019.
>
> [4] Hongyang Zhang, Yaodong Yu, Jiantao Jiao, Eric P. Xing, Laurent El Ghaoui, and Michael I. Jordan. Theoretically principled trade-off between robustness and accuracy. In ICML, 2019a.
>
> [5] Kaiming He, Haoqi Fan, Yuxin Wu, Saining Xie, and Ross Girshick. Momentum contrast for unsupervised visual representation learning. In CVPR, 2020.

---

> > ### Comment · Reviewer_8Qx5 · 2021-08-26
> > **thanks**
> >
> > Thank you for your response!

---

### Official Review · Reviewer_gPjn · 2021-07-13

**Rating:** 5
**Confidence:** 4

**Summary:**

This paper proposes a new method for finetuning pretrained NLP models to achieve both high standard and high robust accuracy. They argue that finetuning on the more difficult and plentiful adversarial examples can lead to catastrophic forgetting, more so than standard finetuning of models. They propose an additional loss wherein the model is encouraged to get high accuracy while also retaining features from the pretrained weights. This improves the accuracy over standard finetuning and various other ways of regularizing adversarial finetuning.

**Limitations And Societal Impact:**

Yes.

**Main Review:**

Strengths:
* I like the motivation and early sections showing that adversarial finetuning shifts the model from its original weights by a much larger amount that standard finetuning.
* There are clear gains across multiple tasks for their method.

Weaknesses:
* My main concern is that the baselines are weak. In particular, the idea of overfitting in adversarial training is well-established  and past work has shown that weight decay, dropout, and these forms of standard regularization are insufficient (https://arxiv.org/abs/2002.11569). Instead of these, early stopping for robust accuracy is the best strategy for improving results. What I would like to see is an early stopping baseline, and what would be even better would be learning curves that show the standard and robust accuracy over time for different finetuning methods. Ideally, the paper's proposed method, if stopped at the proper time, is superior to early stopping using standard finetuning or other regularization strategies.

**Time Spent Reviewing:**

2

---

> ### Author Response · Authors · 2021-08-10
> **Author Response**
>
> We thank the reviewer for the insightful review and valuable feedback.
>
> Regarding the used adversarial baseline, we note that throughout our experiments early stopping is employed for all the compared methods, including the adversarial baseline (please kindly refer to the Appendix line 17-18). As such, our adversarial baseline is strong enough as the reviewer suggested and the proposed method is shown to be superior to all other compared methods. We will move the statement of using early stopping from the Appendix to the main part of the paper to avoid any further confusion and will add a learning curve to the revision. We thank the reviewer for the suggestion and hope your concern is addressed.

---

### Official Review · Reviewer_aWT9 · 2021-07-16

**Rating:** 6
**Confidence:** 3

**Summary:**

This paper first demonstrated that adversarial training does not directly fit a conventional fine-tuning scenario proposed, and then proposed an adversarial fine-tuning method (Robust Informative Fine-Tuning RIFT) from an information-theoretical to fine-tune a pre-trained language model towards robust down-stream performance.

The experimental results on two prevailing NLP tasks (i.e., sentiment analysis and natural language inference) show that RIFT outperforms some baseline fine-tuning methods in terms of robustness.

**Ethics Review Area:**

["Privacy and Security (e.g., consent)"]

**Limitations And Societal Impact:**

Yes, the authors adequately addressed the limitations and potential negative social impact of their work.

**Main Review:**

Clarity:
This paper is clearly written and well organized.

Originality:
This work shows an interesting analysis, that is, the adversarial training fails to retain the generic and robust linguistic features that have already been captured by the pre-trained model. They also proposed an informative adversarial fine-tuning method that maximizes the mutual information between the output of an objective model and that of the pre-trained model conditioned
 on the class label.  The work is inspiring in the robustness of  NLP and easy to follow.

Significance:
 This work is technically sound, and all claims are well supported by both theoretical analysis and experimental results.

Concerns:
- What is the connection between the relative L2 distance (between the pre-trained model and the objective model in the parameter space) and catastrophic forgetting? Does greater L2 distance necessarily mean more forgetting?
- How does RIFT perform on the General Language Understanding Evaluation (GLUE) benchmark?




**Time Spent Reviewing:**

About 2 hour.

---

> ### Author Response · Authors · 2021-08-10
> **Author Response**
>
> We thank the reviewer for the insightful review and valuable feedback.
>
> Regarding the relative L2 distance and forgetting, in our defense, the relative L2 distance is able to show our motivation that adversarial fine-tuning intensifies forgetting. With no explicit regularization on the optimization process to avoid forgetting, chances are low that a larger L2 distance in the parameter space corresponds to a smaller change in the function space. As all methods in Figure 2 do not apply any regularization to avoid forgetting, a greater L2 distance generally means more forgetting in Figure 2. In addition, our motivation can also be justified using downstream tasks performance: the drop in vanilla accuracy of adversarial fine-tuning baseline compared to standard fine-tuning, and encouraging memorization does improve both generalization and robustness in downstream tasks (section 4.5). We thank the reviewer and will add the above discussion to the revision.
>
> Regarding performance on GLUE, as the GLUE test set labels are not publicly available, we are not able to generate the test set adversarial examples for evaluation. As such, we follow previous SOTA defense methods [1, 2] to evaluate the robustness on IMDB and SNLI. We understand that you may want some additional empirical supports and thus we conduct some experiments on the development set of GLUE SST-2. On SST-2, the proposed method also achieves leading robust performance. For example, based on BERT, the proposed method achieves 63.5% accuracy under Genetic attack, surpassing the adversarial baseline by 2.4% (as we are still tuning hyperparameters, the final reported performance of the proposed method on SST-2 may be even higher). We thank the reviewer and will add the results as a supplement to the revision.
>
> \
> [1] Robin Jia, Aditi Raghunathan, Kerem Göksel, and Percy Liang. Certified robustness to adversarial word substitutions. In EMNLP, 2019.
>
> [2] Xinshuai Dong, Anh Tuan Luu, Rongrong Ji, and Hong Liu. Towards robustness against natural language word substitutions. In ICLR, 2021.

---

### Official Review · Reviewer_CwDX · 2021-07-17

**Rating:** 7
**Confidence:** 4

**Summary:**

This paper studies the catastrophic forgetting problem when fine-tuning pre-trained language models towards adversarial robustness. They argue that existing robust training methods, such as adversarial training, make the objective model deviated too much from the pre-trained model, resulting in bad generalization. Starting from an information-theoretical perspective, the authors propose RIFT, a fine-tuning method that maximizes the mutual information between the outputs of the objective model and the outputs of the pre-trained model when conditioned on the labels. The experimental results show that RIFT has a better performance than other baselines, which supports the better generalization as the authors claim. In addition, they give explanations why considering I(S;T|Y) in the objective is better than considering I(S;T).


==== After Rebuttal ====

Thanks for the response. All my questions are properly resolved.

**Limitations And Societal Impact:**

Yes

**Main Review:**

Pros:
- The motivation is clear and convincing.
- The proposed method has a theoretical support.

Cons:
- The improvement of SNLI seems to be small. Providing the significance test may help.


Some questions I have:
- What norm-ball B(x) you consider in the experiments? What's the size of B(x)
- Are the adversarial examples you generate on the embedding-level or token-level? My understanding is that you only find an adversarial perturbation on the embeddings without generating the actual adversarial text.
- Can the proposed method be extended to token-level NLP tasks, such as sequential tagging?


**Time Spent Reviewing:**

5

---

> ### Author Response · Authors · 2021-08-10
> **Author Response**
>
> We thank the reviewer for the insightful review and valuable feedback.
>
> Regarding the B(x) in our experiments, the B(x) we used consists of all the possible combinations of word substitutions using synonyms. We focus on synonyms substitution attacks for robustness evaluation as those attacks are very powerful and yet preserve the semantic and syntactic well. The set of synonyms for each word is defined following previous SOTA defenses against word substitutions [1, 2] for fair comparisons. The size of B(x) differs among datasets: for IMDB, each textual input on average has 108 substitutable words and each word on average has 6 synonyms (thus 6^108 different combinations per input on average); for SNLI, each textual input has 4 substitutable words and each word has 6.5 synonyms on average (6.5^4 different combinations per input on average).
>
> Regarding the relatively small improvement margin on SNLI compared to IMDB, it may relate to the dataset property. Input from SNLI on average has a smaller attack space (6.5^4 combinations per input) compared to IMDB (6^108 combinations per input), and the smaller attack space would to some extent limit the ability of an attack method to differentiate between defenses (the smaller difference between vanilla accuracy and robust accuracy on SNLI compared that on IMDB also supports this point). To further address your concern about significance, we additionally conduct Mann–Whitney U test [5], a nonparametric significance test, on SNLI based on 5 random experiments. For example, on SNLI based on Bert-base-uncased, we examine whether the robust accuracy of the proposed method is significantly greater than that of the runner-up method. The resulting p-value is 0.0425<0.05, showing the statistical significance of the improvement by the proposed method. We thank the reviewer and we will add these results to the revision as suggested.
>
> Regarding the generated adversarial examples, in training, the adversarial examples are modeled using convex hulls on the embedding-level following [2]. During robustness evaluations, actual adversarial texts are generated (by Genetic attacks [3] and PWWS attacks [4]).
> Regarding whether the proposed method can be extended to other tasks, if a task can be formulated as a classification problem, then the proposed method can be applied. For a regression problem, it may necessitate some discretization techniques for the conditional mutual information term and we plan to leave it for future work.
>
> \
> [1] Robin Jia, Aditi Raghunathan, Kerem Göksel, and Percy Liang. Certified robustness to adversarial word substitutions. In EMNLP, 2019.
>
> [2] Xinshuai Dong, Anh Tuan Luu, Rongrong Ji, and Hong Liu. Towards robustness against natural language word substitutions. In ICLR, 2021.
>
> [3] Moustafa Alzantot, Yash Sharma, Ahmed Elgohary, Bo-Jhang Ho, Mani Srivastava, and Kai-Wei Chang. Generating natural language adversarial examples. In EMNLP, 2018.
>
> [4] Shuhuai Ren, Yihe Deng, Kun He, and Wanxiang Che. Generating natural language adversarial examples through probability weighted word saliency. In ACL, 2019
>
> [5] https://docs.scipy.org/doc/scipy/reference/generated/scipy.stats.mannwhitneyu.html

---

> > ### Comment · Reviewer_CwDX · 2021-08-24
> > **All my questions are properly resolved.**
> >
> > Thanks for the response. All my questions are properly resolved.

---

### Decision · Program_Chairs · 2021-09-28

**Decision:**

Accept (Poster)

**Comment:**

This work provides a well-motivated approach to fine-tune a model for robustness while retaining features from pretrained weights. Adversarial robustness in language is an important problem since in general it's trivial for an attacker to slightly change the input. The paper is well-written and the analysis is interesting. The method is well supported theoretically and there is a good set of experiments, however, the improvements in robustness have limited significance while there is still a drop in accuracy on the vanilla evaluation. The rebuttal was well-written and answered the reviewers' questions.

**Consistency Experiment:**

NeurIPS has a long history of experimentation. In 2014, NeurIPS ran an experiment in which 10% of submissions were reviewed by two independent committees to quantify the randomness in the review process. This year, we repeated a variant of this experiment to see how the quality of the review process has changed over time.  This paper was part of the experiment and was therefore assigned to two committees (consisting of reviewers, an Area Chair, and a Senior Area Chair) that reached independent decisions.  If both committees made the same recommendation, this recommendation was followed. If a single committee recommended acceptance, the paper was accepted (with the exception of a few cases in which the other committee identified what we considered a fatal flaw, e.g., an error in a key result).

Both committees reached the same decision: **Accept (Poster)**

The other committee assigned to the paper recommended **Accept (Poster)**.  You can find the other set of reviews, along with any follow up discussion with the authors here:
https://openreview.net/forum?id=pl2WX3riyiq